# Complex Regional Pain Syndrome after Spine Surgery: A Rare Complication in Mini-Invasive Lumbar Spine Surgery: An Updated Comprehensive Review

**DOI:** 10.3390/jcm11247409

**Published:** 2022-12-14

**Authors:** Umberto Aldo Arcidiacono, Daniele Armocida, Alessandro Pesce, Marco Maiotti, Luca Proietti, Giancarlo D’Andrea, Antonio Santoro, Alessandro Frati

**Affiliations:** 1Human Neurosciences Department, Neurosurgery Division, “Sapienza” University, 00185 Rome, Italy; 2IRCCS “Neuromed”, 86077 Pozzilli, Italy; 3Neurosurgery Unit, Santa Maria Goretti Hospital, Via Guido Reni 1, 04100 Latina, Italy; 4Villa Stuart Hospital, Orthopedic Clinic, 00135 Roma, Italy; 5Division of Spinal Surgery, IRCCS Fondazione Policlinico Universitario Agostino Gemelli, Institute of Orthopaedics, Università Cattolica del Sacro Cuore, 00168 Rome, Italy; 6“Spaziani” Hospital, 03100 Frosinone, Italy

**Keywords:** spine surgery, pain, neurosurgery, complex regional pain syndrome, CRPS

## Abstract

Background: Complex regional pain syndrome (CRPS) is a postoperative, misdiagnosed condition highlighted only by pain therapists after numerous failed attempts at pain control by the treating surgeon in the case of prolonged pain after surgery. It only occurs rarely after spine surgery, causing the neurosurgeon’s inappropriate decision to resort to a second surgical treatment. Methods: We performed a systematic review of the literature reporting and analyzing all recognized and reported cases of CRPS in patients undergoing spinal surgery to identify the best diagnostic and therapeutic strategies for this unusual condition. We compare our experience with the cases reported through a review of the literature. Results: We retrieve 20 articles. Most of the papers are clinical cases showing the disorder’s rarity after spine surgery. Most of the time, the syndrome followed uncomplicated lumbar spine surgery involving one segment. The most proposed therapy was chemical sympathectomy and spinal cord stimulation. Conclusion: CRPS is a rare pathology and is rarer after spine surgery. However, it is quite an invalidating disorder. Early therapy and resolution, however, require a rapid diagnosis of the syndrome. In our opinion, since CRPS occurs relatively rarely following spinal surgery, it should not have a substantial impact on the indications for and timing of these operations. Therefore, it is essential to diagnose this rare occurrence and treat it promptly and appropriately.

## 1. Introduction

Prolonged postoperative pain is the most common post-surgical complication in spine surgery and often represents a significant limitation in the patient’s functional recovery and prolonged hospitalization time [1]. Spine surgery today is considered a low-risk surgery with side effects that tend to be transient and self-limiting [2], and the introduction of minimally invasive spine surgery further reduced postoperative pain [3]. Rarely, a frustrating and intense form of postoperative pain could appear unresponsive to the traditional analgesic medications, causing the neurosurgeon’s inappropriate decision to resort to a second surgical treatment that could lead to a higher rate of complications, prolonged hospitalization, and costs. In such cases, the hypothesis of a disorder condition may be considered complex regional pain syndrome (CRPS).

CRPS is a challenging orthopedic, neurological, and traumatic condition [4] that occurs following an extraneous trauma, such as surgery, fractures (1–2%), peripheral nerve damage (2–5%), and infections. CRPS is an undiagnosed condition represented by a constellation of symptomatology; the continuing pain, disproportionate to any trigger event, is the main feature [5], associated with autonomic dysfunction, swelling, dystrophic skin changes, and stiffness, functional impairment, and atrophy [5]. Physiopathology is complex, and a combination of different factors can occur at the time of initial injury, leading to nervous system sensitization, autonomic dysfunction, and inflammatory changes [6]. In 1995, the International Association for the Study of Pain (IASP) [7] defined this term to define and diagnose this condition. CRPS is an often-unknown condition highlighted only by pain therapists after numerous failed attempts at pain control by the treating surgeon.

The IASP definition notes, “following an initial and spontaneous pain event…, the spread of pain is not limited to the affected nerve area…” [7]; this makes recognizing this pathology more arduous when incurring after spinal surgery, as degenerative spine pathology with chronic pain and ganglionar damage often result in spreading pain over the entire area of nerve radiation [8,9,10,11].

So, the true incidence of CRPS and subsequent treatment in spinal pathology of orthopedic and neurosurgical interests are unknown [12,13]. 

We performed a review of the literature reporting and analyzing all recognized cases of CRPS in patients undergoing minimally invasive spinal surgery, intending to identify the best diagnostic and therapeutic strategies for this unusual and intriguing syndrome. In this study, we also bring to attention a patient who developed a diagnosed CRPS after a surgical procedure without complications. We compare our experience with the cases reported through a literature review.

## 2. Materials and Methods

The study was conducted following Cochrane database and PRISMA recommendations [14] for systematic reviews. The English literature was systematically investigated using MEDLINE, the NIH Library, and Google Scholar. The last search date was 28 February 2021. Search terms included: “Complex regional syndrome” or “CRPS” in combination with “spine surgery”.

Searches were limited to human studies and English languages, but there were no limits regarding the period of publication. Backward citation tracking was applied to identify articles not retrieved by electronic searches.

### Selection Criteria

Two independent authors conducted the selection of abstracts for full review based on predefined inclusion and exclusion criteria. Studies were eligible if they reported original data on all cases concerning CRPS after spine surgery as the main topic. To identify relevant articles, we screened the titles and abstracts for keywords regarding spine surgery and post-surgical complications. We also manually searched and cross-referenced the reference lists of relevant articles to identify additional studies that were not detected through the initial literature search. We also decided to include in our research “Reflex sympathetic dystrophy” since it was the previous name of the disorder before the introduction of the Budapest criteria.

Studies were excluded if they reported language other than English, presented a re-analysis, commentary or review summarizing the results of the previous series or if the topic was not centered on the CRPS after surgery. Each author reviewed the abstracts independently and generated a list of studies to retrieve for full-text review.

## 3. Results

### 3.1. Data Extraction

The research returned a total of 145 papers on CRPS as a topic. To this initial cohort, the exclusion mentioned above criteria were applied, selecting 21 papers for the analysis. After the study selection, we further excluded 10 papers where the clinical information about the onset of pain syndrome was missing or the CRPS was not linked to the surgical procedure. The present paper of the senior author’s personal experience (A. F.) was added. The final cohort is composed of 10 papers. After applying selection criteria, we analyze data for 72 patients containing the CRPS reported data as a possible complication after spine surgery with clinical information. The remaining patient information is derived from our institutional experience.

A data extraction sheet was prospectively designed to extract all of the necessary information visible in Figure 1. The two authors subsequently verified the data. The following details were extracted: the number of patients described, the level on which the surgery was performed, the treatment after the CRPS diagnosis, and the timing, if described, of the onset and the resolution (Figure 1).

As we can see from the table, most of the papers are clinical cases showing the disorder’s rarity after spine surgery. Most of the time, the syndrome followed uncomplicated lumbar spine surgery involving one segment. Only two case reports have been found to report the pathology after cervical spine surgery (3.5%). In 60/72 (83.3%) patients, the hypothesized cause was nerve compression during the surgical procedure. In all reported cases, the use of NSAIDS with cortisone (in which the dosage is never specified) was reported as the initial treatment. Only in 18 patients, including ours, is the time of resolution or improvement of symptoms reported. In the case of medical treatment failure, the most adjuvant therapy used was chemical sympathectomy and spinal cord stimulation. Statistical analysis could not be performed due to the exiguity of the sample.

### 3.2. Representative Case

A 61-year-old man, with clinical history of colon cancer 5 years before, was admitted to our department after complaining about low back pain with irradiation along the left leg and dysesthesia of L5 dermatome lasting for 6 months. Magnetic resonance imaging (MRI) examination revealed the presence of L5-S1 lumbar disk herniation with instability (retrolisthesis, Modic type I. Figure 2A,B). The patient was treated with anterior and posterior arthrodesis of L5-S1 using transforaminal lumbar interbody fusion using the minimally invasive spine surgery (MIS-TLIF) technique and L5-S1 discectomy (Figure 2C,D). The procedure and postoperative course were uneventful and prompt discharge occurred. Ten days after the discharge, the patient reported a new onset of paresthesia and dysesthesia on the back of the left foot along with the territory of the L5 nerve. 

The initial symptoms were change in progressive pain, particularly low-back pain with irradiation of the left leg with night exacerbation. The patient started treatment with NSAID, pregabalin, and myorelaxant, with low pain control. Due to the relentless and worsening of the pain, a new lumbar MRI was performed, showing the presence of recurrent L5-S1 paramedian disk herniation (Figure 2C,D) and a new surgical intervention was performed. 

The surgery is performed using Wiltse’s transmuscular approach [4]. The old bar is removed, and the left laminectomy is extended on L5 and S1. Under the operating microscope, a small left paramedian disc fragment is recognized. It is in contact with the S1 root and so it is removed. Microsurgical lysis of adhesions, fibrotic tissue and newly formed inflammatory vessels are performed. The left L5 root appears swollen and distended and particularly inflamed. The conjugation foramen and the left L5-S1 lateral recess is wholly opened and released. At the end of the procedure, the L5 (at the level of foramen) and S1 (in the recess) roots appear completely free and well-mobilized; there are no evidence of compression or adhesions. The bar is repositioned, the screw nuts are changed and tightened with dynamometric system.

#### Postoperative Period

On the second postoperative day, the patient complains about the same symptomatology of the admission. Opioid drugs are administered, and the pain is controlled. The day after, a new MRI is performed, normal surgical outcomes are shown and fluid collection (washing and blood) is present. The fluid collection is then sucked in sterile manner and then the patient is discharged with good control of the pain with NSAID. The subsequent course after discharge is ordinary and no pain is referred.

Ten days after the procedure, the patient complains of edema and redness of the left leg and foot (Figure 3), severe pain disproportionate to any inciting event (ex. light touch). The ecocolordoppler excludes vascular pathology. A neurological evaluation is considered, and according to “Budapest Criteria” [5], the diagnosis of CRPS formulated: the presence of allodynia, temperature asymmetry, and skin color changing along the left distal lower limb. 

The patient is treated with amitriptyline, pregabalin and deltacortene [7,8]. With these therapies, there is a continuous slow improvement of painful symptoms with a marked reduction in edema of the left foot, partial recovery of sensitivity, improvement in walking. Foot pain almost resolves during the day; painful nocturnal access is still present, which, however, presents a faster resolution after opioids intake. After the dismission, the therapy continues for 2 months and physiotherapy begins for 6 months. The patient undergoes lumbar MRI at 30 days and a control neurosurgical clinic. Seven days following the return home, the patient begins physical rehabilitation and after 6 months, reported a complete resolution of the symptomatology.

## 4. Discussion

In 1995, the International Association for the Study of Pain (IASP) [7] coined the term CRPS to exactly define the condition where, following an initial and spontaneous pain event or allodynia/hyperalgesia, the spread of pain is not limited to the affected nerve area, and the intensity is not related to the initial event [4,7]. Therefore, modified diagnostic criteria (‘Budapest criteria’) were proposed and validated recently [5]. Actually, CRPS can be categorized into Types 1 and 2: the two different subtypes differentiate according to the presence of a major nerve damage (CRPS Type 2) or the absence of a major nerve damage (CRPS Type 1). A third subtype, identified as CRPS-NOS, denotes a clinical picture consistent with CRPS but does not meet the full clinical diagnostic criteria, with no other diagnosis more appropriate [5,15].

CRPS is a disorder that can arise following trauma [16,17], whether accidental or surgical. However, it is not mentioned in the literature as a typical postoperative complication of spine surgery [18,19]. Indeed, our literature search confirms that CRPS, especially after spinal surgery, is reported very little in the literature, but we also believe that this is partly due to the lack of knowledge of this condition and its relative misdiagnosis. Knoeller et al. [20] theorized a reaction to “instrumental mobilization” as an etiological cause of CRPS after spine surgery. During a MS-TLIF approach, it is unknown if the small surgical corridor, the ganglion and sympathetic trunk could be traumatized. Cheng et al. [21] proposed this theory as a response to the work of Morr et al. [22]. The suggested critique was on the improper positioning of the dilatation tube that could have damaged the nerve. We know that the sinuvertebral nerve takes a recurrent course, re-entering the spinal canal through the intervertebral foramen via the deep anterior intraforaminal ligament, lying alongside the pedicle cephalad to the corresponding disc. It joins the somatic root from the ventral ramus and an autonomic root provided by the grey ramus. It arises bilaterally from the ventral ramus of each spinal nerve just distal to the dorsal root ganglia, supplying both proprioceptive and nociceptive fibers [23].

For this reason, a more lateral approach such as that of Wiltse, although considered less traumatic [24], may have a greater risk of traumatizing the sinuvertebral nerve and indirectly creating inflammation along its course and the autonomic root. Most of the time, among the 72 patients considered, the approach used is often not mentioned. There is no scientific evidence [18,19] that the different type of approach to the spine determines a different incidence or prevalence of CRPS, leading to this being a speculative theory.

A noxious stimulus usually elicits an innate immune response by releasing inflammatory peptides such as tumor necrosis factor-α (TNF-α), interleukin 1β, and interleukin 6. During continuous acute and progressive pain in CRPS, there is a massive release of inflammatory cytokine. This condition leads to increased axon reflex vasodilation in the affected extremity [4], causing increased inflammatory protein extravasation. This phenomenon is commonly called “neurogenic inflammation”.

Moreover, besides peripheral neurogenic inflammation, another phenomenon occurs: neuroinflammation [6,25]. Neuroinflammation refers to inflammation occurring within the nervous system due to glial cell activation, leading to the increased production of pro-inflammatory cytokines and chemokines [26]. The peripheral sensitization and the neuroinflammation lead to a central sensitization involving remodeling in the spinal cord, somatosensory cortex, ventromedial frontal cortex and right anterior part of the insula [4,27]. 

The etiopathology of the syndrome is complex and requires multiple collaborating elements. Therefore, the hypothesis is that the movement between the segments is somehow associated with inciting a painful stimulus [28,29]. After the initial triggering event, several interconnected macroscopic and microscopic changes in the disc occur in degenerative spinal disease. It begins with microscopic damage leading to inflammation. The degenerating disc releases growth factors, such as bFGF and TGF-beta1, increased tumor necrosis factor-α, interleukin-1β, Nerve Growth Factor, NGF, and Brain-Derived Neutrophic Factor, BDNF [30]. Chronic trauma (such as chronic degenerative lumbar pathology) can be the initial cause of the inflammatory cascade and the increased neuronal activity of the primary afferent nerve. A continuous inflammatory stimulus leads to the onset of neuroinflammation, which is responsible for the transition from acute to chronic pain and the maintenance of chronic pain in CRPS [6,25]. 

Considering our patients’ cohort, most of them have a history of chronic low back pain; the primum-movens that induced the production of inflammatory cytokines, and then the surgical stimulus may have accentuated the onset of the syndrome. Therefore, the great nociceptive stimulus together with the surgical stimulus induced a hyperinflammatory reaction that, in a predisposed subject, may have induced the onset of CRPS after a trauma, such as surgery. We recognized a more common onset of the CRPS in patients with chronic lumbar pathology than those with a cervical one (Table 1).

Among the objectives of our work is the research of anamnestic elements suggestive of expecting a probable postoperative CRPS. However, due to the rarity of the pathology, the few cases reported, and the scarce details available from the various case reports or series, we cannot establish any predictive factor. The onset time and the time of recovery are vary greatly. Symptoms usually resolve in mean time between 2 and 40 days. 

It is revealed from the review that in degenerative lumbar pathology [16,31,38,39], the relationship with the onset and causes of CRPS is more complex. In fact, many authors initially stated that the pain syndrome could be related to nerve, ganglion, or nerve root compression or impingement [38,39], but this was followed in the majority of cases (71/72) by spontaneous or only pharmacological resolution, without recourse to a second surgery. CRPS is always a diagnosis of exclusion, after proper evaluation of a well-positioned surgical procedure and implant, but it should be considered among the hypotheses from the outset in order to avoid the risk of an unnecessary second surgery.

While the most classic presentation of CRPS [5,15] is a lasting disease that does not resolve and is managed symptomatically with limited success, this is not the rule for CRPS as a postoperative spine surgery complication. An effective therapy proposed is the sympathetic chemical block; however, prompt recognition of the syndrome allowed for a rapid commencement of specific pharmacological therapy and, therefore, a complication resolution [22].

A multidisciplinary approach should be a standard practice in the treatment. The literature reports that conservative modalities such as rehabilitation and psychotherapy play a role. Mental disorders such as the onset of depression and anxiety impact the prognosis, worsening it [29]. 

Classical medical therapies include the use of Bifosphonate, free-radical scavengers (Dimethyl sulfoxide 50% (DMSO) and N- 435 acetylcysteine (NAC), steroid (prednisolone), antidepressants, and anticonvulsants (gabapentin) [40,41,42]. 

Steroid therapy is considered the drug of choice in CRPS treatment, usually prednisolone. There is no clear consensus on the optimal dosing and treatment duration; the usual dose recommended is from 30 mg/day up to 200 mg/day tapered over several weeks [28,30].

The importance of initiating CRPS and early therapy has been pointed out frequently [15] and seems associated with a less severe prognosis and rapid symptom resolution. Early therapy, however, requires a prompt diagnosis of the disorder. However, it is critical to be aware of the possibility and the knowledge of CRPS onset to promote an early diagnosis and more successful therapies.

### Limitations and Further Study

The main limitation of this analysis is the paucity of information available from the cases reported in the literature about the surgical procedure, onset of symptoms, timing, and outcome, which prevents a true critical analysis. Moreover, since CRPS has been recently described and systematized, many cases may have been erroneously reported with other definitions, not identifiable by a systematic research approach. The condition, although rare, can be frustrating for most treating physicians, and in the future it will be necessary to supplement this case report with new case descriptions and details of treatment and outcome as is highlighted in our case report.

## 5. Conclusions

CRPS is a rare pathology and is rarer after spine surgery. However, being a quite invalidating disease, early therapy requires a rapid diagnosis of the disorder. In our opinion, CRPS should not have a substantial impact on the indications for and timing of these operations. However, it is critical to be aware of the possibility that CRPS may occur in particular in a patient with a history of depression, previous trauma and re-intervention. In spinal surgery, one of the main postoperative symptoms is postoperative pain, which can occur in the first days of the surgical procedure, increase the days of hospitalization, delay mobilization, and the beginning of rehabilitation. Therefore, it is essential to diagnose this rare occurrence and treat it promptly and appropriately. 

## Figures and Tables

**Figure 1 jcm-11-07409-f001:**
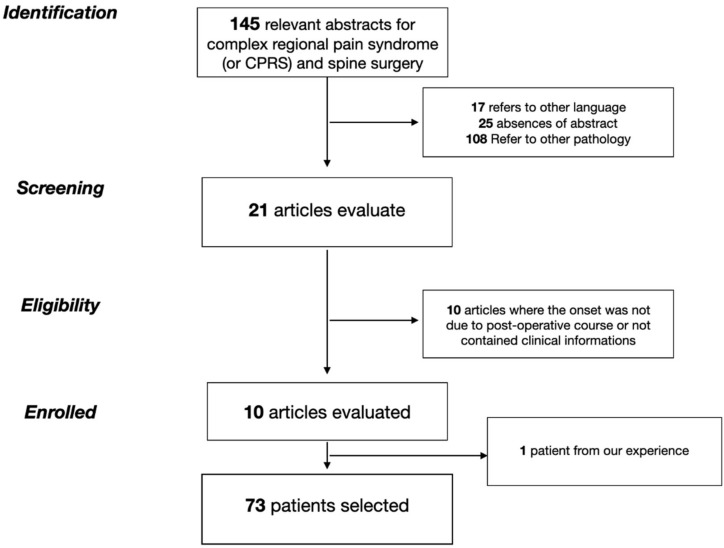
Patient selection flow-chart.

**Figure 2 jcm-11-07409-f002:**
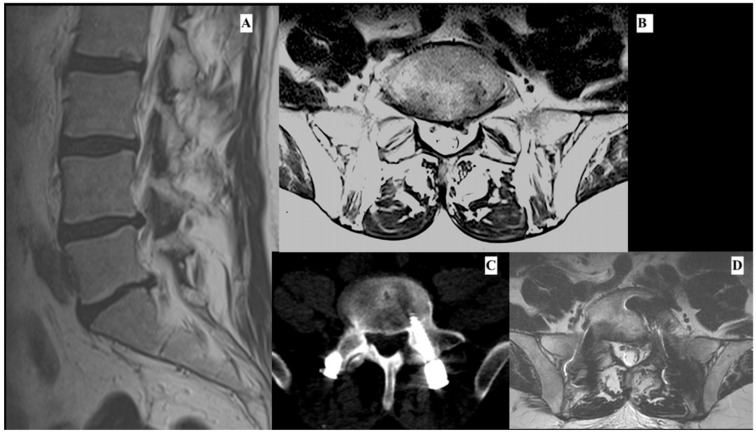
This shows a sagittal (**A**) and axial (**B**) imaging of preoperative lumbar MRI. It is possible to see a paramedian-intraforminal left lumbar disk hernia. The image shows an axial slice of postoperative lumbar TC (**C**). It is possible to observe the correct positioning of the screw. Image (**D**) shows an axial slice of lumbar MRI, which was performed 1 month after the first surgery due to the continuous intractable pain. It is possible to observe at level L5-S1 a paramedian left lumbar disk hernia (**D**).

**Figure 3 jcm-11-07409-f003:**
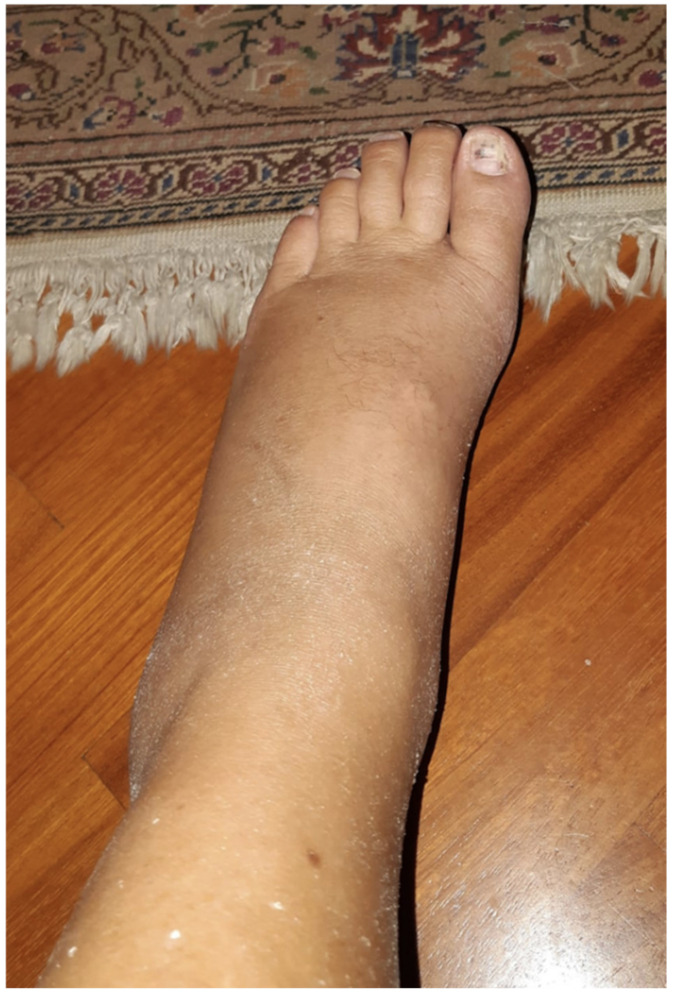
Figure shows a picture of the leg of the patient. It appears swelled and red.

**Table 1 jcm-11-07409-t001:** Patient’s selection.

Study	n. Patients	Level	Treatment	Probably Cause	Outcome
Carlson et al., 1977 [28]	2	L4-L5	Not informed	CRPS due to nerve compression	
Bernini et al., 1981 [31]	1	L4-L5	Sympathetic Block	CRPS due to nerve compression	
Chodoroff et al., 1985 [32]	1	L5-S1	Surgical Decompression	CRPS due to nerve compression	
De Weerdt et al., 1987 [33]	48	(26) L4-L5 (25) L5-S1	Not informed	CRPS due to nerve compression	
Mockus et al., 1987 [34]	12	Not specified	Surgical Sympathectomy	Onset not due surgery	92% improvement after 1 month
Ballard et al., 1991 [35]	1	L4-L5	Laminectomy and Discectomy	CRPS due to nerve compression	No detail on resolution
Perrot et al., 1992 [29]	4	L5-S1	Nerve release (Neurolysis)	CRPS due to nerve compression	Improvement in 1–3 months
Adachi et al., 1994 [36]	1	L4-L5	lumbar sympathetic block	CRPS due to nerve compression	
Condon et al., 1998 [37]	1	L5-S1	Discectomy + Surgical Sympathectomy	CRPS due to nerve compression	
Se Hee Kim et al., 2016 [22]	1	L4-L5	percutaneous nucleoplasty	CRPS due to nerve compression	Resolution after procedure
Our case	1	L5-S1	Discectomy + Lumbar spine stabilization	Not known	Improvement after 2 months

## Data Availability

Not applicable.

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
