# Peer review of "Complex Regional Pain Syndrome after Spine Surgery: A Rare Complication in Mini-Invasive Lumbar Spine Surgery: An Updated Comprehensive Review"

_jcm, 2022, doi:10.3390/jcm11247409_

Round 1

Reviewer 1 Report

This review article is well written. It looks at CRPS after spinal surgery. It will add to the current literature. 

The abstract is well structured. 

The introduction is detailed.

The methodology is well described, the inclusion is detailed.

The results is written well with examples.

The discussion is complete and comprehensive.

The conclusion concludes the article well.

Not much needs revising. 

Author Response

Thank you so much for your appreciations.

Reviewer 2 Report

In this paper, the authors performed a systematic review of the literature reporting and analyzing all recognized and reported cases of CRPS in patients undergoing spinal surgery to identify the best diagnostic and therapeutic strategies for this unusual condition. As the authors indicated, since CRPS occurs relatively rarely following spinal surgery, it is essential to diagnose this rare occurrence and treat it promptly and appropriately in case of inappropriate second surgical treatment. The methods are well designed and results are convincing. There was only one question, where is the table 1? I think the paper would be more strengthened if the author provided more detail about the 72 patients containing the CRPS and comment more in discussion about the 10 papers included.

Author Response

Thank you for the review and your appreciation. We believe that due to a loading error the table with the flow-chart of study selection was not viewable, we have included it in the text and discussed in more detail the information transposed from the included studies.

Reviewer 3 Report

1. What is the diagnostic criteria of CRPS? What is the characteristic of CRPS after spine surgery when compared to the general CRPS? Please clarify it in introduction part.

2. In the “3.2. Representative case”, whether an additional MRI was performed at ten days after the second procedure. Whether the possible symptom of “CRPS” was correlated with the colon cancer, because the patient had a clinical history of colon cancer. If not, please state it.

3. The postoperative nerve injury is a common complication of spine surgery. In the “discussion” part, how did you differ the Type 2 CRPS (with the presence of a major nerve damage) with the postoperative complication. Some chronic diseases (e.g., diabetes) could also cause nerve injuries, whether CRPS had a higher incidence in these groups of patients.

4. Please add several updated studies ( 1.Cervical dorsal root ganglion stimulation for complex regional pain syndrome: Technical description and results of seven cases; 2.Parameters of Spinal Cord Stimulation in Complex Regional Pain Syndrome: Systematic Review and Meta-analysis of Randomized Controlled Trials; 3.Efficacy and Safety of Cervical and High-Thoracic Dorsal Root Ganglion Stimulation Therapy for Complex Regional Pain Syndrome of the Upper Extremities)

Author Response

Response to the reviewer 3:

Thank you for the careful review of our work, unfortunately as we received the decision of the 3 reviewer after the submission of the revised version we altered the bare minimum without altering the confidence given in the re-submission. However we have responded point-to-point the suggested advice and added the requested references:

What is the diagnostic criteria of CRPS? What is the characteristic of CRPS after spine surgery when compared to the general CRPS? Please  clarify it in introduction part.

R: We better clarified this point in the introduction section: In the IASP definition where, “following an initial and spontaneous pain event, the spread of pain is not limited to the affected nerve area”[7], makes recognizing this pathology more arduous when incurring after spinal surgery, as degenerative spine pathology with chronic pain and ganglionar damage often result in spreading pain over the entire area of nerve radiation.

  1. In the “3.2. Representative case”, whether an additional MRI was performed at ten days after the second procedure. Whether the possible symptom of “CRPS” was correlated with the colon cancer, because the patient had a clinical history of colon cancer. If not, please state it.

R: The patient had been pre-operatively studied in general and was already in radiological and clinical follow-up for colon cancer so a manifestation of disease without colic symptoms was immediately ruled out from oncologist.

  1. The postoperative nerve injury is a common complication of spine surgery. In the “discussion” part, how did you differ the Type 2 CRPS (with the presence of a major nerve damage) with the postoperative complication. Some chronic diseases (e.g., diabetes) could also cause nerve injuries, whether CRPS had a higher incidence in these groups of patients.

R: Normal post-operative complication of nerve injury occurs early in patients undergoing spine injury and does not reappear 10 days after treatment without other symptoms. Of course, safety often does not exist since moderate analgesic therapy is always set in the immediate postoperative phase. With this question, the reviewer has fully centered the great difficulty of diagnosis and management of the rare postoperative CRPS that we hope the very publication of this paper will recognize.

  1. Please add several updated studies ( 1.Cervical dorsal root ganglion stimulation for complex regional pain syndrome: Technical description and results of seven cases; 2.Parameters of Spinal Cord Stimulation in Complex Regional Pain Syndrome: Systematic Review and Meta-analysis of Randomized Controlled Trials; 3.Efficacy and Safety of Cervical and High-Thoracic Dorsal Root Ganglion Stimulation Therapy for Complex Regional Pain Syndrome of the Upper Extremities)

R: We added and discuss it.

Please address all correspondence concerning this manuscript to me at

[email protected]

Thank you for your consideration of this manuscript.

Best regards,
